Machine learning classification of mango maturity based on carotene content from Raman spectra

Tan Ji Loun 1
Hashim Fazida Hanim fazida@ukm.edu.my 1 2
Sampe Jahariah 3
Baseri Huddin Aqilah 1
Salim Ghassan Maan 1
Md Ali Sawal Hamid 1
1 Department of Electrical, Electronic and Systems Engineering, Faculty of Engineering and Built Environment, Universiti Kebangsaan Malaysia , Bangi , Selangor , Malaysia
2 Research Centre for Sustainable Process Technology (CESPRO), Faculty of Engineering and Built Environment, Universiti Kebangsaan Malaysia , Bangi , Selangor , Malaysia
3 Institute of Microengineering and Nanoelectronics (Institut Kejuruteraan Mikro dan Nanoelektronik), Universiti Kebangsaan Malaysia , Bangi , Selangor , Malaysia
Omara Timothy
Electronic publication date: 2025 Nov 18
Publication date: 2025
Volume: 13
Electronic Location ID: e20288
Received 2025 May 5; Accepted 2025 Oct 3
Copyright: ©2025 Tan et al.
Copyright year: 2025
Copyright holder: Tan et al.
License: This is an open access article distributed under the terms of the Creative Commons Attribution License, which permits unrestricted use, distribution, reproduction and adaptation in any medium and for any purpose provided that it is properly attributed. For attribution, the original author(s), title, publication source (PeerJ) and either DOI or URL of the article must be cited.
License URL: https://creativecommons.org/licenses/by/4.0/

Keywords: Mango, Ripeness, Raman spectroscopy, Carotene, Machine learning

Funding: Universiti Kebangsaan Malaysia through the Research University Grant GUP-2022-027 Faculty of Engineering and Built Environment, Universiti Kebangsaan Malaysia The Article Processing Charge (APC) for this publication was supported by Universiti Kebangsaan Malaysia through the Research University Grant (GUP-2022-027) and the Faculty of Engineering and Built Environment, Universiti Kebangsaan Malaysia. The funders had no role in study design, data collection and analysis, decision to publish, or preparation of the manuscript.

==============================
Determining mango ripeness is essential for ensuring its delicious taste, enticing aroma, and rich nutritional value. For farmers, harvesting mangoes too early can result in stunted fruit and lower yields compared to those harvested at a ripe stage. This study aims to develop a potentially non-invasive and efficient method for detecting mango ripeness using Raman spectroscopy. Traditional methods, which rely on human assessment and color evaluation with image processing, are inconsistent, inaccurate, and time-consuming due to variations in mango color and individual differences in vision and perception. To address these limitations, this study pursued three main objectives: extracting data characteristics of organic compounds in mangoes based on raw Raman spectrum data, identifying the correlation between carotene characteristics and mango ripeness levels, and evaluating the performance of machine learning models in classifying mango ripeness levels. A total of  29 mango fruit spectra were analyzed, with 13 samples selected to represent three ripeness categories: underripe, ripe, and overripe. Raman spectra peak signal analysis revealed that mango peel contains lycopene, β-carotene, lutein, and neoxanthin, all of which are derived from carotenoid molecules in the range of 1,480 cm−1 to 1,550 cm−1. Statistical analysis confirmed the significance (p < 0.05) of extracted Raman Peak Intensity features in distinguishing ripeness levels, supported by high correlation coefficients between carotenoid peak intensity and mango maturity. This study achieved 100% accuracy in classifying mango ripeness levels using three classifier models: the Medium Gaussian Support Vector Machine, the Cubic Support Vector Machine, and the Weighted K-Nearest Neighbors. Raman spectroscopy has proven to be a reliable and robust method, immune to external factors such as light, humidity, and noise, which makes it a promising approach for assessing mango ripeness.

Introduction

Mango is one of the most esteemed tropical fruits globally, renowned for its delicious taste, enticing aroma, and rich nutritional value (Lebaka et al., 2021; Zhang, Zhu & Zhu, 2022). In the 21st century, mangoes emerged as the popular tropical fruit in terms of production, with a remarkable global output exceeding 51 million tons in 2019 (Lebaka et al., 2021; Islam et al., 2017). Malaysia primarily produces mangoes for the local market and exports them to other countries, such as Singapore, Hong Kong, and Brunei (Ding & Darduri, 2013). Malaysia’s mango production industry has gained international recognition not only for its large production volumes but also for being a good source of vitamin (Rozana, Suntharalingam & Othman, 2017; Uda et al., 2020). In addition to its delicious taste, mango has a rich chemical composition, including structural carbohydrates such as pectin and cellulose, and major amino acids like lysine, leucine, cysteine, valine, arginine, phenylalanine, and methionine (Lebaka et al., 2021; Maldonado-Celis et al., 2019; Mustafa et al., 2023). During the ripe stage, the unsaturated fatty acids & lipid composition increase, showing significant levels of omega-3 and omega-6 fatty acids (Maldonado-Celis et al., 2019; Yahia et al., 2023; Hernández-Estrada et al., 2022). The primary pigments in mango are chlorophylls (a and b) and carotenoids, while the main organic acids are malic acid and citric acid (Maldonado-Celis et al., 2019; Deshpande et al., 2016). The unique aroma of mango comes from a diverse group of volatile components that undergo significant biochemical, physiological, and structural changes during development and ripening, affecting its nutritional and phytochemical composition, as well as its aroma, taste, and antioxidant capacity.

For consumers, ripe mangoes offer the best balance of taste, texture, and nutritional value (Appiah, Kumah & Idun, 2011). Unripe mangoes are complex and less sweet, while overripe mangoes are prone to rot and may breed harmful microorganisms (Lebaka et al., 2021). For farmers, the timing of mango harvesting is essential, as harvesting too early results in stunted fruit and lower yields (Kour et al., 2018). Therefore, understanding mango ripeness can lead to informed harvesting and marketing decisions, ultimately increasing farmers’ income and operational efficiency. Traditional methods for determining mango ripeness rely on visual assessment of color, which are prone to inconsistencies due to variations in mango color and individual differences in vision and perception (Zulkifli et al., 2018). The researchers have explored image processing methods for ripeness assessment, but these methods suffer from accuracy issues due to changes in light intensity throughout the day and under different lighting conditions (Zulkifli et al., 2018). To address these limitations, this study pursued three main objectives: extracting data characteristics of organic compounds in mangoes based on raw Raman spectrum data, identifying the correlation between carotene characteristics and mango ripeness levels, and evaluating the performance of machine learning models in classifying mango ripeness levels. The novelty of this study lies in its integration of raw Raman spectral analysis with carotenoid identification and advanced machine learning classification for mango ripeness detection. Unlike prior works that focus solely on surface color or basic spectral fingerprints, this study explores the distinct vibrational signatures of ripeness-related compounds which are carotenoids as main features in supervised learning classification. Moreover, this work contributes spectral dataset collected under real-world conditions and evaluates multiple machine learning models to determine the optimal approach for robust and non-destructive mango ripeness classification.

Raman spectroscopy is one of the vibrational analysis techniques in food chemistry, which is valued for its non-destructive approach, high sensitivity, and capability to provide straightforward interpretation and precise structural identification of both organic and inorganic compounds without extensive sample preparation (Sun et al., 2022; Tzuan et al., 2022; Raj et al., 2021; Rostron, Gaber & Gaber, 2016). According to Dan et al. (2018), each raw Raman spectrum consists of signals convolved and contributed by different organic compounds (Dan et al., 2018). Notably, Raman spectroscopy enables in situ analysis to determine structural composition under field conditions. Its applications extend beyond assessing mango ripeness to include monitoring plant health, early disease diagnosis, and identifying biotic and abiotic structures in plants. A recent study by Trebolazabala et al. (2017) successfully employed portable Raman spectroscopy to investigate changes in carotene and chlorophyll a levels in tomatoes at different ripening stages, demonstrating the potential for establishing an accurate automated grading system (Trebolazabala et al., 2017). By focusing on carotenoids as Raman scatterers, the accuracy of ripeness assessments can be significantly improved. Developing machine learning models to identify key indicators of ripeness for automated ripeness classification systems has also been proven in previous research (Tzuan et al., 2022; Raj et al., 2021).

Furthermore, a comparative analysis of fruit ripeness detection technologies identified three key competitors: computer vision, near-infrared spectroscopy (NIR), and Raman spectroscopy (Zulkifli et al., 2018; Tzuan et al., 2022; Raj et al., 2021; Makky, Soni & Salokhe, 2014; Silalahi et al., 2016). While computer vision leverages optics, image processing, and pattern recognition, its reliance on RGB values for color analysis can lead to inaccuracies due to varying lighting conditions. Although NIR spectroscopy is non-destructive, it requires frequent instrument calibration, which can extend measurement time and is essential for obtaining accurate results. In contrast, Raman spectroscopy has emerged as the superior choice, demonstrating excellent performance in accurate and non-invasive ripeness assessment.

Materials & Methods

This section describes the materials and methods used in this study, beginning with the preparation of mango fruit samples, the Raman spectroscopy setup for acquiring raw Raman spectra, spectral processing for feature extraction, statistical analysis to select the optimal features, and finally, the classification analysis tools employed. This study comprises seven main stages as illustrated in Fig. 1, each systematically designed to achieve the research objectives. These stages include the collection and preparation of mango fruit samples, acquisition of raw Raman spectral data, Raman spectral pre-processing for noise reduction and feature extraction, correlation and statistical analysis for data interpretation, and classification analysis to determine mango ripeness using appropriate machine learning techniques.

Figure 1 Research framework for machine learning-based classification of mango maturity using Raman spectroscopy.

Mango fruits sample preparation

In this study, 29 mango fruit samples were collected from private plantation trees and purchased from a local grocer. The mangoes were classified into three ripeness categories (underripe, ripe, and overripe) based on color, texture, and taste (Uda et al., 2020). Carotenoids are pigments that produce bright yellow, red, and orange colors in plants, vegetables, and fruits. They are present in both the skin and pulp of mangoes (Lebaka et al., 2021). Measuring carotenoids in fruit pulp can be challenging due to the presence of other pigments such as chlorophyll and anthocyanins. Additionally, as chlorophyll content decreases, the carotenoid concentration in the pulp and peel of the fruit will increase (Lebaka et al., 2021; Maldonado-Celis et al., 2019; Yahia et al., 2023).

Hence, exocarp samples were taken from each fruit, representing the top (T), middle (M), and bottom (B) parts, to account for color variations influenced by sunlight exposure during growth. Following the collection of mango skin samples, each section was carefully mounted onto glass microscope slides and analyzed using a Raman spectrometer to obtain the corresponding raw spectral data, as shown in Fig. 2.

After obtaining the Raman spectrum in the form of a CSV file, each spectrum is processed using Orange Data Mining and OriginPro 2017 software to obtain hidden peaks resulting from molecular vibrations in the mango skin sample. In this step, mango species that do not have a clear peak and are not suitable for this study will also be discarded.

Raman spectroscopy

Raman spectroscopic measurements of mango samples with varying ripeness levels, including underripe, ripe, and overripe fruits, were conducted using the Thermo Scientific™ DXR™ 2xi Raman spectroscopy system (Thermo Fisher Scientific, Waltham, MA, USA). This advanced system enables point measurements at submicron spatial resolution across a large area. To ensure accurate results, a 532 nm green laser was employed, as it aligns with the Raman equation I ∝ 1/λ4, where I represents the Raman vibration intensity and is inversely proportional to the wavelength, λ4.

Figure 2 Raman spectroscopy analysis on mango samples at different maturity stage.

To maintain the structural and molecular composition integrity of the mango fruit’s interior, careful control of the sample was important. The 532 nm laser was directed to the sliced skin of the mango fruit to obtain a microscopic view. From this view, the area to be scanned was selected, and a spot-sized local measurement was conducted. Subsequently, the spectroscopy system calculated and determined the average value for the selected region, generating an output spectrum that comprised Raman intensity and wave number. This Raman spectrum analysis allowed for the accurate categorization of mango ripeness levels. By studying the distinct Raman spectra associated with underripe, ripe, and overripe mango samples, the methodology aimed to establish a reliable and effective approach for classifying mango ripeness.

Raman spectral processing

Data pre-processing in this study involves three key stages: background noise removal, data smoothing, and spectral segmentation. These stages are crucial for subsequent feature extraction processes. The primary objective of data pre-processing is to effectively eliminate noise signals, ensuring smoother and more accurate analysis. Additionally, it aims to transform raw data into a format suitable for machine learning procedures. Noise in the raw data may arise from instrumentation or environmental factors, making data pre-processing crucial in maintaining data quality and addressing the challenges inherent in the raw data.

To address fluorescence contamination in Raman spectra, the rubber band algorithm was employed for background noise removal. This step is crucial for controlling noise data before proceeding with feature extraction. Consequently, a Savitzky-Golay digital filter was utilized for data smoothing.

Furthermore, a spectral segmentation process was applied to the smoothed Raman spectrum, allowing for the elimination of irrelevant portions that are not pertinent to this study. The primary focus is on the vibrational band located in the wavelength range of 1,480–1,550 cm−1. By concentrating on this specific part of the spectrum, the analysis can target key features associated with mango ripeness levels, thereby facilitating accurate classification.

Following spectral segmentation, the data underwent a deconvolution process utilizing curve-fitting techniques. In this step, a sum of four Lorentzian profiles was employed, and the profiles were manually positioned to establish the initial curve-fitting parameters. The resulting deconvoluted spectra exhibited variations in peak positions, which were subsequently calibrated using the pure β-carotene peak position as a reference marker.

Statistical analysis

Statistical analysis is employed to examine the characteristics extracted from the data, which helps to identify features with low statistical significance and exclude them from the dataset. The homogeneity test is utilized to determine variations in variance between the three ripeness levels. If variance is detected, follow-up tests, namely the Brown-Forsythe and Welch tests, are performed to validate the results. Additionally, a one-way ANOVA is conducted to analyze the minimum differences in selected characteristics across ripeness levels and provide insights into features with low statistical significance.

Subsequently, a multiple comparison analysis was performed as a post hoc test to determine the exact minimum differences between the three categories (underripe, ripe, and overripe) for the characteristics that passed the ANOVA test. In this study, Gabriel’s test is chosen due to the non-uniform sizes of the mango fruit samples. The Games-Howell test is also conducted to confirm the characteristics further.

The significant characteristics that undergo statistical analysis include the position of the Raman peak, the intensity of the Raman peak, the full-width half-maximum (FWHM), and the intensity ratio of the Raman peak. Features showing statistical significance were selected for the subsequent classification analysis. This comprehensive approach ensures accurate classification of mangoes based on their ripeness levels.

Classification analysis

In this study, machine learning techniques were employed to analyze the ripeness level of mangoes using the Classification Learner function in MATLAB software. Specifically, two machine learning algorithms, such as support vector machine (SVM) and K-Nearest Neighbors (KNN), were utilized for classification purposes.

The essential features obtained from the statistical analysis process were carefully selected as input data for the machine learning models. By leveraging these significant features, the machine learning classifiers can effectively learn patterns and relationships that distinguish between different ripeness levels of mangoes.

The primary evaluation metric used was accuracy, defined as the ratio of correctly classified instances to the total number of cases. Accuracy was chosen for its simplicity and direct interpretability in scenarios with balanced class distributions where the mango samples were relatively evenly distributed across the ripeness categories (underripe, ripe, and overripe).

Furthermore, confusion matrix analysis was used to visualize and interpret model performance across all ripeness classes, allowing for the identification of any systematic biases or misclassification trends. Additionally, receiver operating characteristic (ROC) curves were used to evaluate the discriminative ability of the models for binary classification subproblems.

Results

In this section, the focus was on a comprehensive examination of various aspects related to the mango fruit samples collected for this study. Specifically, we conduct a detailed analysis of mango fruit spectra, examine the distinctive features present in the data, perform statistical analysis to uncover patterns and trends, and showcase the outcomes of the classifier analysis.

After comparing the average Raman spectra of the top, middle, and bottom regions of 29 mangoes, the results showed that local mango varieties, such as Padi Mango and Chok Anan Mango, and foreign varieties, such as R2E2 Mango, exhibited clear and distinct peaks. This may be attributed to the absence of food-grade paraffin wax on the surfaces of these mangoes, unlike other varieties. The 29 samples comprised Mango R2E2 (19), Mango Susu (2), Mango EV (1), Mango Golden Lily (1), Mango Aiwen (2), Mango Chok Anan (1), and Mango Padi (3). However, only 13 R2E2 mangoes were included in the final analysis due to inconsistencies or low signal quality in the Raman spectra of the excluded samples. The selected sample size was determined based on data quality considerations, given the strong spectral consistency and significant variance observed across the retained samples. Food-grade paraffin wax can interfere with the light collection process used in Raman spectroscopy and cause changes in the resulting spectrum, which consists of distinct chemical compositions and can affect the Raman spectra (Prinsloo, du Plooy & van der Merwe, 2004).

In addition, the selection of 13 Mango R2E2 samples (five unripe, four ripe, and four overripe) for this study was based on their popularity and wide accessibility across diverse retail settings, including supermarkets, grocery stores, and fruit markets.

Mango fruits Raman spectra

Among the five peaks observed in the Raman spectra of R2E2 Mango, peak 5 recorded the highest Raman intensity value, while peak 1 had the lowest Raman intensity value. The sequence of peaks, based on increasing Raman intensity values, is as follows: peak 1, 4, 2, 3, and 5. Raman bands are related to the molecular vibrations present in mango fruit samples. Additionally, the peak position of the Raman spectrum is connected to the structure of the functional groups. Furthermore, the observed Raman intensity value for each peak provides information about molecular density; higher molecular density results in lower peak intensity, whereas lower molecular density leads to higher peak intensity. It is essential to study the relationship between the peak position in the Raman spectrum and the material’s structure.

The Raman spectrum obtained in this study showed that each observed peak corresponds to molecular vibrations in the carotenoid compound, as shown in Fig. 3. In addition, Fig. 4 also shows the difference in Raman spectra between underripe, ripe, and overripe mangoes. These compounds consist of molecular chains, such as C=C, C-C, CH3, and CH2, each producing a distinct vibrational mode, and verify the findings from Raj et al. (2021). The first peak, located around 956 cm−1, results from molecular vibrations of C-C, CH2, and CH3 in carotenoids. The second peak at 1,005 cm−1 is attributed to the C-H stretching vibration of the CH3 group. Moreover, the third peak found at 1,150 cm−1 is associated with C-C molecular vibrations. Additionally, the fourth peak, located at around 1,270 cm−1, is possibly representing C-H molecular vibrations. Lastly, the fifth Raman peak, situated at about 1,515 cm−1, is attributed to the vibration of the C=C molecule.

According to Trebolazabala et al. (2017), ripe and overripe fruits exhibited high levels of β-carotene. β-carotene is responsible for the red-orange pigment found in fruits, vegetables, and trees (Trebolazabala et al., 2017). As a result, the presence of β-carotene is less noticeable in underripe fruits. The lack of β-carotene content at the less mature stage can also be attributed to the possible dominance of the green pigment contributed by chlorophyll a compounds. Therefore, the selection of the 1,480 cm−1 to 1,550 cm−1 range for this study was motivated by the presence of β-carotene in this particular range.

Figure 3 Raman spectra of mango fruit with molecular vibration mode references.

Figure 4 Difference in Raman spectra between underripe, ripe and overripe mango.

The Raman peak position documented in this study aligns with findings reported by Raj et al. (2021). Peak 5 observed in the Raman spectrum originates from various organic compounds, including β-carotene, lycopene, lutein, and neoxanthin (Raj et al., 2021).

In this study, all four hidden peaks in Peak 5, denoted as V1, V2, V3, and V4, have been successfully identified, each with a unique peak position, as shown in Fig. 5. Based on theoretical analysis, the V2 Raman band located around 1,516 cm−1 is associated with β-carotene molecules. The V1 Raman band that is positioned to the left of the V2 Raman band can be attributed to the lycopene molecule. The justification for this interpretation is that linear molecules, such as lycopene, have a lower peak position compared to β-carotene molecules, due to the linear structure of lycopene and the ring structures at both ends of β-carotene (Maoka, 2020).

In addition, the V3 and V4 Raman bands, positioned more to the right than the V2 Raman band, are interpreted as carotenoid molecules with oxygenated rings in their structures. According to the study by Ruban et al. (2001), the carotenoid molecule most closely related to β-carotene, with a difference of four to six cm−1, is known as lutein (Ruban et al., 2001). Consequently, the V3 Raman band is attributed to the lutein molecule. Lastly, the V4 Raman band exhibits a peak at around 1,530 cm−1, attributed to Neoxanthin molecules. This finding aligns with the research reported by both Ruban et al. (2001) and Raj et al. (2021).

Figure 5 The results of the deconvolution process of the Raman spectrum of mango fruit by using the Lorentz curve adjustment method.

Feature analysis

The features extracted in this study are Raman peak position, Raman peak intensity, Raman peak intensity ratio, and Raman peak full width at half maximum (FWHM). Additionally, the results of feature extraction will be compared with those of previous studies, and further correlations will be established.

The peak position is a critical characteristic (also known as a chemical fingerprint) that is important for determining the molecular properties of carotenoids in mangoes. The mean values of Raman peak positions extracted from a range of 1,480 cm−1 to 1,550 cm−1 are illustrated in Fig. 6A. This plot demonstrates the distribution of peak positions for three levels of mango ripeness: underripe (URP), ripe (RP), and overripe (ORP). Four distinct bands labeled as V1, V2, V3, and V4 represent different organic compounds found in mango skin, with V1 corresponding to lycopene, V2 to β-carotene, V3 to lutein, and V4 to neoxanthin. The significance of Raman peak positions lies in their ability to act as chemical fingerprints for the organic compounds/molecules present in mangoes. This information enables the identification of specific organic compounds associated with each Raman peak, facilitating the determination of the corresponding vibrational mode of the molecule. The variation in the position of the Raman peak signifies the change in the molecular composition of carotenoids as mangoes progress from underripe to ripe and overripe stages. In this case, there is a high variation in V1 and V4 bands for underripe mangoes, which suggests the potential presence of other organic compounds.

The peak intensity observed in the Raman analysis serves as an indicator of the concentration of different carotenoid molecules present in the mango fruit sample, as shown in Fig. 6B. During the underripe stage, the V3 band exhibits the highest mean intensity among all bands, indicating a relatively higher concentration of lutein. The mean intensities of the V1 and V4 bands are closer to each other, suggesting a lower concentration. The ascending order of average peak intensity is as follows: V4, V1, V2, V3. As the mango fruit reaches the ripe stage, the peak intensity of the V2 band displays the highest mean intensity compared to the other bands. This intensity has increased by 1,186% compared to the previous level, indicating a substantial rise in the concentration of β-carotene. Additionally, the intensities of the V1, V3, and V4 bands also show an increase, with the lowest mean intensity still observed for the V1 band. The order of ascending peak intensity is as follows: V1, V2, V3, V4. At the overripe level, the intensity of the V2 band still reaches the highest mean value compared to the other bands, indicating a continued high concentration of β-carotene. On the other hand, the V4 band exhibits the lowest mean intensity, suggesting a lower concentration of neoxanthin. The order of ascending peak intensity for the overripe stage is as follows: V4, V1, V3, V2.

The third feature is derived from the second feature, which is the intensity ratio of the Raman band. This ratio represents the relative concentrations of different molecules present, as indicated by the ratio of peak intensities in Raman spectroscopy. To establish the ratios, the peak intensity of the V2 band (β-carotene) was used as a reference value. Three ratios were generated: V2/V1, V2/V3, and V2/V4. The first ratio represents the concentration ratio between β-carotene and lycopene molecules. The second ratio represents the concentration ratio between β-carotene and lutein molecules. Lastly, the third ratio represents the concentration ratio between β-carotene and neoxanthin molecules. Since each carotenoid molecule exhibits a unique color, these ratios also reflect the color distribution over the mango skin. Figure 6C illustrates the differences in the V2/V1, V2/V3, and V2/V4 peak intensity ratios for the three ripeness levels. In both the underripe and ripe mango stages, the V2/V1 peak ratio shows the highest value among all studied ratios. This indicates a higher concentration of β-carotene molecules compared to lycopene molecules during these stages. However, when mangoes reach the overripe stage, the V2/V1 ratio decreases due to a higher concentration of lycopene (a red pigment) compared to β-carotene. A similar trend is observed in the V2/V4 and V3/V4 peak ratios compared to the V2/V1 ratio. These trends in peak ratios offer valuable insights into the changes in carotenoid composition in mangoes at various ripeness levels. Additionally, they shed light on the changing molecular properties and dynamics of pigmentation as the fruit ripens.

Figure 6D illustrates the changes in FWHM for all the bands of mangoes at three maturity levels: underripe, ripe, and overripe. During the underripe stage, the V1 band displayed the highest mean FWHM value, indicating a broader spectral line. In contrast, the V4 band showed the lowest mean FWHM value, suggesting a narrower spectral line. At the ripe stage, the V2 band exhibited the highest mean FWHM value, indicating a broader spectral line. In contrast, the V1 band recorded the lowest mean FWHM value, implying a narrower spectral line. At the overripe stage, the V2 band still recorded the highest mean FWHM value, while the V4 band showed the lowest mean FWHM value, indicating a narrower spectral line. Furthermore, the mean FWHM values of the V2 and V4 bands exhibited similar trends during the three ripeness levels, showing consistent changes in their spectral line widths as mangoes progressed from underripe to ripe and then to overripe. This information provides valuable insights into the molecular interactions and structural changes occurring in carotenoid compounds as mangoes ripen. The FWHM analysis contributes to a better understanding of the ripening process and may aid in assessing mango fruit quality based on spectral characteristics.

Statistical analysis

In this study, a total of 15 Raman spectrum features were extracted from four hidden Raman peaks obtained through the deconvolution process using Lorentz curve fitting from Peak 5. The importance of these features was determined using various statistical tests, including the Homogeneity test, One-way ANOVA test, Gabriel’s test, Welch’s test, and Games-Howell’s test. These tests aimed to identify and retain only the most essential features while excluding those of less significance. All statistical analyses were conducted using IBM SPSS Statistics software. The identified important characteristics will serve as crucial predictor variables in the subsequent training process of the classification model used to assess different ripeness levels of mango fruit.

After performing the statistical analysis, it was observed that all 15 features passed both the ANOVA test and the Welch test, as shown in Fig. 7. Statistical analysis confirmed the significance (p < 0.05) of extracted Raman Peak Intensity features in distinguishing ripeness levels, supported by high correlation coefficients between carotenoid peak intensity and mango maturity. This indicates that each of these extracted Raman spectral features exhibits significant variation across different ripeness levels of mango fruit. Consequently, all features are considered valuable and informative for further analysis and machine learning classification.

Figure 6 Feature analysis of Raman spectra.

(A) Mean plot of Raman peak position for 3 levels of ripeness (B) Mean plot of Raman peak intensity for 3 levels of ripeness (C) Mean plot of Raman peak intensity ratio for 3 levels of ripeness (D) Mean plot of Raman peak FWHM for 3 levels of ripeness (URP, underripe; RP, ripe; ORP, overripe).

As a result, this study proposes the use of four predictors: the Raman peak intensity of V1, V2, V3, and V4 in the classifier analysis. The ANOVA test revealed a significant level below 0.05 for these predictors, as shown in Fig. 7, indicating the reliability and strong evidence supporting the results. These predictors are derived from the critical Raman spectral features, which are expected to play a vital role in accurately classifying and determining the maturity level of mangoes.

Figure 7 Significant characteristics identified through statistical analysis ANOVA test.

Classification analysis

The classification analysis utilized the four predictors, namely the intensity of the V1 to V4 peak bands, as described in the previous section. This analysis was evaluated using two types of classification algorithms with different kernel types: KNN and SVM, as listed in Table 1. The number of mangoes in the training set for this study is small, which may result in overfitting. Overfitting can lead to high accuracy in the training set validation but low accuracy when testing external mango samples. Therefore, cross-validation was employed to evaluate model performance, using a 4-fold (k = 4) validation strategy to ensure reliability and mitigate overfitting due to the small dataset.

Discussion

In this study, we conducted a comprehensive analysis of Raman spectra obtained from mango fruit samples at various ripeness levels to understand the molecular composition and changes in carotenoid compounds during the ripening process. Our study revealed five distinct Raman peaks, each corresponding to specific molecular vibrations in the carotenoid compound. Peak 5 exhibited the highest Raman intensity value, while peak 1 had the lowest intensity; the sequence of peaks, based on increasing intensity values, was peak 1, 4, 2, 3, and 5.

Table 1 Results of classification analysis using four predictors in MATLAB software.

Classifier type	Kernel methods	Accuracy (%)	
SVM	Linear	92.3	
	Quadratic	92.3	
	Cubic	100.0	
	Fine Gaussian	76.9	
	Medium Gaussian	100.0	
KNN	Fine	92.3	
	Medium	23.1	
	Coarse	38.5	
	Cosine	30.8	
	Cubic	30.8	
	Weighted	100.0	

The Raman peak positions proved to be critical characteristics for identifying the molecular properties of carotenoids in mangoes. Through deconvolution using Lorentz curve fitting, we identified four prominent bands labeled as V1, V2, V3, and V4, which represented different organic compounds in mangoes: V1 (lycopene), V2 (β-carotene), V3 (lutein), and V4 (neoxanthin). These peak positions served as chemical fingerprints, enabling the identification of specific organic compounds associated with each Raman peak and aiding in the determination of the corresponding vibrational modes.

In addition, an intriguing finding was the relationship between Raman peak intensity and molecular density in mango fruit samples. This correlation provided valuable insights into the molecular composition and concentration of carotenoids in different ripeness levels of mangoes. Our analysis demonstrated that mangoes progress from underripe to ripe and then to overripe stages, resulting in changes in the molecular composition of carotenoids.

Moreover, the variation in Raman peak intensities provided essential information about the concentration of different carotenoid molecules in mango fruit samples. During the underripe stage, the V3 band exhibited the highest mean intensity, indicating a relatively higher concentration of lutein. As the mango fruit reached the ripe stage, the V2 band displayed the highest mean intensity, indicating a substantial rise in the concentration of β-carotene. These intensity trends continued in the overripe stage, with the V2 band still recording the highest mean intensity, indicating a continued high concentration of β-carotene.

Furthermore, the derived peak intensity ratios serve as indicators of molecular concentrations in mango fruit samples. The V2/V1, V2/V3, and V2/V4 ratios reflected the relative quantities of β-carotene compared to lycopene, lutein, and neoxanthin, respectively. These ratios also provided insights into the color distribution over the mango fruit skin, as carotenoid molecules exhibit distinct colors.

Additionally, our analysis of full width at half maximum (FWHM) values sheds light on molecular interactions and structural changes in carotenoid compounds as mangoes ripen. The FWHM analysis contributed to a better understanding of the ripening process and may serve as a basis for assessing mango fruit quality based on spectral characteristics.

The statistical analysis of the extracted Raman spectral features confirmed the significance of all 15 features in differentiating mango fruit maturity levels. Subsequently, we employed classification algorithms to assess the ripeness level of mango fruits based on the extracted features. The classification models demonstrated remarkable accuracy with Medium Gaussian SVM, Cubic SVM, and Weighted KNN achieving a maximum accuracy of 100%. This high accuracy validates the effectiveness of the selected peak bands (V1 to V4) as reliable indicators for accurately determining mango ripeness. This classifier has been compared with previous studies listed in Table 2, and each study provides valuable insights.

Table 2 Comparison of the results of the classifier model and the method from previous studies.

Classifier method	Detection technique	Accuracy (%)	Reference	
ANN	Imaging (Ripeness)	93.50	Makky, Soni & Salokhe (2014)	
ANN	Image Processing	94.00	Alfatni et al. (2014)	
ANN	HIS Color Model	70.00	Shabdin et al. (2016)	
KNN	Raman spectroscopy	100.00	Raj et al. (2021)	
ANN	Raman spectroscopy	97.90	Tzuan et al. (2022)	
KNN/SVM	Raman spectroscopy	100.00	This study (2025)	

In terms of classification performance, the machine learning models developed in this study which include Medium Gaussian SVM, Cubic SVM, and Weighted KNN achieved a perfect 100% accuracy. This result surpasses previous imaging-based approaches using artificial neural networks (ANN), such as Makky, Soni & Salokhe (2014) and Alfatni et al. (2014), which reported accuracies of 93.5% and 94%, respectively, and Shabdin et al. (2016), which achieved 70% using an HIS color model. These vision-based methods might be limited by their sensitivity to lighting and color variation. Compared to previous Raman spectroscopy-based studies, Tzuan et al. (2022) achieved 97.9% accuracy using ANN, while Raj et al. (2021) achieved 100% using KNN. However, this study uniquely combines both SVM and KNN with carotenoid selected peak bands (V1 to V4), optimized preprocessing and robust classification models. This study not only for precise molecular insight into ripeness indicators like carotenoids but also for real-time application in automated fruit sorting where provides both theoretical advancement and practical significance in the field of non-destructive fruit quality analysis.

However, several practical limitations must be acknowledged. Firstly, while Raman spectroscopy is a powerful tool in controlled environments, its implementation in field conditions is challenged by ambient light interference, fruit surface variability, and potential equipment sensitivity. Additionally, although the current dataset provides strong classification accuracy, the sample size and variety may not fully represent the heterogeneity of mangoes across different regions and cultivars. In addition, another limitation is the assumption of uniformity in the fruit’s surface chemistry. Mangoes may exhibit localized variations in carotenoid concentration due to microclimatic growth factors, which could influence spectral readings if not properly sampled and accounted for.

In summary, our study offers valuable insights into the molecular composition and changes in carotenoid compounds at various ripeness levels of mangoes. The Raman peaks, peak positions, intensities, and ratios serve as essential characteristics for understanding mango fruit ripening and assessing mango fruit quality. These findings hold significant implications for the mango industry and open up opportunities for non-destructive ripeness assessment techniques and quality control measures. However, further research is warranted to explore additional aspects and applications of Raman spectroscopy in the field of fruit analysis. Raman spectroscopy can be integrated with portable scanners and automated systems, where a Reinforcement Learning (RL) agent intelligently selects representative mangoes for scanning based on visual cues, reducing the need to monitor every fruit. This innovative sampling approach enables efficient, large-scale maturity detection in real-world farm or post-harvest settings while maintaining non-destructive assessment. Our results also demonstrate that Raman spectroscopy is a beneficial analytical technique in the field of food chemistry.

Conclusions

This study successfully addressed three key objectives related to the analysis of mango ripeness using Raman spectroscopy and machine learning. The identification of specific Raman bands, particularly those associated with C=C double bond vibrations, provided critical molecular markers for evaluating ripeness. The established correlation between carotenoid attributes and ripening stages formed the basis of a non-destructive and reliable assessment technique. Through optimized feature selection and the application of Medium Gaussian SVM, Cubic SVM, and Weighted KNN, the study achieved a classification accuracy of 100%, surpassing the performance of previous imaging-based approaches and demonstrating the robustness of Raman-based classifiers.

These numerical results affirm the practical applicability of the method for rapid and accurate mango maturity detection. However, potential limitations include variability that may be due to sample heterogeneity, environmental interference during spectral acquisition, and the need for standardization across the sample. To enhance the applicability of this method, future research should incorporate larger, multi-regional datasets, investigate real-time field deployment using portable Raman devices, and integrate automatic spectral deconvolution algorithms. Validating the framework in commercial and post-harvest environments will further solidify its value for the agricultural sector. The integration of Raman spectroscopy with machine learning presents a promising approach for scalable, accurate, and non-invasive fruit quality assessment in real-world applications.

Supplemental Information

Supplemental Information 1 Raw Data for All Mango Data

Raman spectra from 29 mango samples, with measurements taken from the top, middle, and bottom parts of each mango. These data were used for Raman spectral processing, statistical analysis, and classification analysis.

The authors gratefully acknowledge the research support from UKM-YSD Chair for Sustainability.

Additional Information and Declarations

Competing Interests

Author Contributions

Data Availability

The authors declare there are no competing interests.

Ji Loun Tan conceived and designed the experiments, performed the experiments, analyzed the data, prepared figures and/or tables, authored or reviewed drafts of the article, and approved the final draft.

Fazida Hanim Hashim conceived and designed the experiments, performed the experiments, analyzed the data, prepared figures and/or tables, authored or reviewed drafts of the article, supervision and Project administration, and approved the final draft.

Jahariah Sampe performed the experiments, authored or reviewed drafts of the article, and approved the final draft.

Aqilah Baseri Huddin performed the experiments, analyzed the data, authored or reviewed drafts of the article, and approved the final draft.

Ghassan Maan Salim authored or reviewed drafts of the article, and approved the final draft.

Sawal Hamid Md Ali analyzed the data, authored or reviewed drafts of the article, and approved the final draft.

The following information was supplied regarding data availability:

The raw measurements are available in the Supplementary File.

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
