# Peer review of "Machine learning classification of mango maturity based on carotene content from Raman spectra"

_PeerJ, doi:10.7717/peerj.20288_

## Round 0.1 · original submission · Major Revisions

· Academic Editor

Major Revisions

Dear authors,

Reviewers have recommended the reconsideration of your manuscript after substantial revision.

We look forward to recieving your revised manuscript.

Best regards,

Timothy Omara

**Language Note:** The review process has identified that the English language must be improved. PeerJ can provide language editing services - please contact us at [email protected] for pricing (be sure to provide your manuscript number and title). Alternatively, you should make your own arrangements to improve the language quality and provide details in your response letter. – PeerJ Staff

·

Basic reporting

Writing can be make more precise (Example: First 2 paragraphs of Introduction can be accommodate in 1 paragraph)

In Introduction answer of "What is Raman Spectroscopy?" is expected.

Can include dataset samples (Images - at least 3 of each category)

Photograph of Instrument used is expected to get clear idea of experimental setup.

Experiment and analysis required more samples (29 & 13 are less for conclusion)

I have one doubt. If you can explain it in paper it would be great.
As we are expecting non-destructive method for ripeness detection. Also you mentioned in Introduction you want to help farmer to know the maturity before picking up.
My doubt is, how Raman Spectroscopy can be utilised for mass mango maturity detection? Usually no of mangoes from a farm will be in thousands. I want answer in the context of real world testing not for research. How it will be implemented practically in farm (or post harvest places)

Experimental design

Expect more clearity in writing. Expect more samples in experiment.

Validity of the findings

No comment

Additional comments

There are many classifiers nowadays. You can test your results with multiple classifier and multiple kernels.

·

Basic reporting

1. The abstract lacks key numerical results. Please include the main findings related to: Peak signal analysis from Raman spectra, Statistical analysis outcomes, and Classification model performance (e.g., accuracy, precision)
2. The application of machine learning for fruit maturity or ripeness detection is not discussed. Please incorporate a review of previous studies that used ML models for this purpose, and clearly highlight the novelty of applying K-Nearest Neighbors (KNN) and Support Vector Machine (SVM) in this context.
3. Revise the methodology and results sections to use past tense, reflecting completed actions. Phrases such as “is” and “will be” should be avoided in this context.

Experimental design

4. The manuscript states that 29 mango samples were obtained from both private plantations and local grocers, but does not specify the number from each source. Please provide this breakdown and justify the sample size, especially with respect to statistical significance and model training sufficiency.
5. Clarify the criteria used during collection to ensure balanced representation across underripe, ripe, and overripe categories. Include how these categories were determined or classified (e.g., color, firmness, Brix level).
6. Specify the version of OriginPro software used for spectral analysis, as this affects reproducibility.
7. The current explanation of the ML pipeline is insufficient. Please provide a clear flowchart or schematic showing all steps from: Data collection, Preprocessing, Feature selection, Model training, and Evaluation. This is essential for reproducibility.
8. Justify the selection of KNN and SVM over other models (e.g., decision trees, random forest, ANN), including why they are appropriate for this specific dataset or application.

Validity of the findings

9. The discussion refers to different mango varieties (e.g., Padi Mango, Chok Anan, R2E2), but these were not introduced in the methodology. Please clarify which varieties were included, how many samples were collected for each, and their role in the classification.
10. Clearly explain how the accuracy metric was used to evaluate the performance of the machine learning models, and provide justification for its selection over other metrics such as precision, recall, F1-score, confusion matrix analysis, or ROC/AUC, if applicable. In addition, it is recommended to include at least one or two additional performance metrics to provide a more comprehensive evaluation of the model's effectiveness.
11. The sentence “cross-validation is required to be used to evaluate performance” (lines 370–371) is ambiguous. Please clarify whether this is a recommendation or a result, and provide details on the type of cross-validation used (e.g., k-fold, LOOCV).
12. Explain why the classification models were trained using only four predictors. Could model performance be improved with additional features (e.g., spectral bands, morphological traits)?
13. The discussion section is insufficiently developed. Expand the discussion for each major result, particularly: Comparisons with previous studies, Interpretation of model performance, Possible limitations or sources of error

Additional comments

14. The conclusion is too general and should be revised to briefly highlight key numerical findings, practical implications, and study limitations. Avoid mentioning objectives and repeating methods, and include clear recommendations for future research.

---

## Round 0.2 · Minor Revisions

· Academic Editor

Minor Revisions

Dear Authors,

Thank you for submitting the revised manuscript.

The revision improved the draft, and I now invite you to address the minor revisions requested by the reviewers. For the results to be scientifically robust and meaningful, readers need to appreciate how your findings align with, diverge from, or advance existing knowledge.

We look forward to receiving your revised manuscript

With kind regards,

Timothy Omara

·

Basic reporting

The novelty and key contributions of the study are not clearly stated in the introduction. Please revise this section to explicitly highlight what distinguishes this work from existing studies and what new insights it provides.

Experimental design

no comment

Validity of the findings

The discussion of the results is currently descriptive and lacks comparative analysis with previous studies. Strengthen this section by critically comparing your findings with relevant literature to contextualize your results and support your conclusions.

Additional comments

no comment

---

## Round 0.3 · accepted · Accept

· Academic Editor

Accept

Administratively accepting this again to complete the shift from AChem to PeerJ.